# Pathways of Transition to Organic Agriculture in Morocco

Hamza El Ghmari [1,2], Rachid Harbouze [1] and Hamid El Bilali [2,*]

1    Hassan II Institute of Agronomy and Veterinary Sciences, Rabat P.O. Box 6202, Morocco
2    International Centre for Advanced Mediterranean Agronomic Studies (CIHEAM-Bari), 70010 Bari, Italy
*    Correspondence: elbilali@iamb.it

**Abstract:** Agriculture is a vital sector in Morocco through its contribution to the gross domestic product (13%) and workforce (30%). However, the sector faces important sustainability challenges due to Morocco's dependence on rain-fed crops for 90% of the production and its vulnerability to climate change. In this context, organic agriculture presents itself as a promising alternative to valorize production and mitigate climate change effects. This article describes the dynamics and development of the organic agriculture niche in Morocco through the lens of the Multi-Level Perspective (MLP) on socio-technical transitions. The MLP is a widely used framework that bases its analysis on transitions being the result of the interaction of niches, socio-technical regimes, and socio-technical landscape. Results of the literature review and semi-structured interviews show that, although the organic niche is relatively well established (11,000 ha of organic land area in 2019), it is still developing at a slower rate than expected due to multiple setbacks. While organic farming does solve many sustainability challenges that Moroccan agriculture faces, it still lacks the infrastructure and human capital to succeed as a niche. All in all, organic farming is still in the first transition stages and can follow a multitude of pathways before becoming relevant in the current agri-food system.

**Keywords:** sustainability transition; multi-level perspective; organic farming; agri-food system; Morocco

## 1. Introduction

In the past twenty years, the Moroccan economy has experienced remarkable growth. This economic prosperity has translated to demographic development, which, in turn, has increased the pressure on Morocco's natural resources [1]. During this period the Moroccan government recognized the importance of agriculture and started multiple action plans to support and modernize its agricultural sector. The Green Morocco Plan (GMP) was the most famous one of these programs; starting in 2008 and lasting 12 years, it had a general objective of improving the agricultural production of the country [2–4].

Thus, agriculture went through a significant evolution with the modernization of practices and the intensification of production in selected areas where farm areas and resources were more abundant, while other areas remained traditionally managed with little inputs [5]. This situation has created a divide in the Moroccan agricultural system, where small farmers produce to survive and big firms invest in intensive production systems to reach competitive yields [6].

Currently, Morocco is facing the outcome of years of unsustainable uses of resources as well as climate change's negative effects on the environment such as irregular rainfall, droughts, degradation of soil, desertification, and pollution [7–9]. This situation has made dealing with sustainability challenges difficult, as the governmental programs have to consider different production systems at once while trying to uphold the system's sustainability [10,11].

The complexity of such challenges does not allow solving them using incremental solutions that optimize one aspect at a time. Instead, they require radical solutions in the form of sustainability transitions, which change the entire system into a more sustainable

one [12–14]. Sustainability transitions are deep changes that alter the configuration of socio-technical systems to adapt to the grand environmental and social challenges that humanity faces [6,7]. The main fields of interest in transition research were the energy and mobility systems, but, later on, studies on transitions in the agri-food system were starting to become more popular [15]. Today, sustainability transitions in the agri-food system are a crucial component in facing food security and food sustainability challenges [16]. Due to their complex and multi-dimensional nature, these challenges are adequately addressed by the frameworks of transition research, such as the Multi-Level Perspective (MLP), which is why they gained enough legitimacy to become part of the agenda of many initiatives in the agri-food arena [17,18].

The transition towards organic agriculture is an example of such transitions that have the potential to enhance the environmental and socio-economic performances [19] of the Moroccan agri-food system [20–22]. Organic agriculture can be defined as "a production system that sustains the health of soils, ecosystems, and people. It relies on ecological processes, biodiversity and cycles adapted to local conditions, rather than the use of inputs with adverse effects. Organic agriculture combines tradition, innovation, and science to benefit the shared environment and promote fair relationships and good quality of life for all involved" [23]. Organic farming systems can help build flexible food systems as they encourage diverse cropping systems in a single farm [24]. In the Moroccan context, organic farming has the potential to provide a better living for farmers while answering the environmental challenges in a sustainable way [25–28].

In this context, this article aims to show the existing research gap in the topic, while answering the following questions: what are the different components at play in this transition (i.e., niche, regime, and landscape) and what pathway is the transition likely to follow?

## 2. Materials and Methods

In the present article, we will be gathering both primary data through semi-structured interviews and secondary data from the literature, and analyzing them through the MLP framework in order to describe the type and features related to the pathway of transition to organic agriculture in Morocco and to define the environment components, internal and external factors, that affect the organic transition and understanding the relations between them.

To establish the validity of our data, we relied on triangulation to confirm whether the data we collected through interviews and the literature review was valid [29,30].

After checking the validity of data through triangulation, we proceed to the presentation of the results through the lens of the MLP framework. In order to describe accurately the organic transition in Morocco, we first described the state of the three socio-technical levels (niche, regime, and landscape) of the agri-food system and focused on the characterization of the niche's development stage. Secondly, we will analyze the different interactions between the three levels to define the niche's position relative to the existing regime. Lastly, we will combine elements from these results to identify the niche's development stage and its interaction with the regime and landscape and, finally, we will define the type of pathway the transition is likely to follow according to Geel and Schot's typology [31].

### 2.1. Analytical Framework

One of the starting points in the field of transition research is the fact that society faces grand environmental challenges that are related to consumption and production patterns. These patterns and the systems that uphold them cannot be altered by following incremental change, instead they require deeper changes or "sustainability transitions" in order to transform them into more sustainable systems. Therefore, transition studies aim to conceptualize and understand how these changes occur and how they affect society's functions [12].

To understand transitions occurring in society, we can conceptualize services that society offers as socio-technical systems. These systems consist of actors, institutions alongside their underlying regulations and norms, and material artefacts and knowledge. These elements' interaction is what provides specific services for society. A transition is a deep change that involves multiple dimensions: technological, material, organizational, institutional, political, economic, and socio-cultural. Additionally, it involves a wide range of actors and takes place in considerable time spans. During this transition, products, services, and business models can be altered through replacement or complementation. The transition also affects technological and institutional structures as well as consumers' perception of the service that is undergoing the transition [32].

Sustainability transitions can be described as the process that pushes established socio-technical systems to shift towards a more sustainable configuration of production and consumption. As mentioned before, this transition is multi-dimensional and takes a long time, but, most importantly, it adheres to the concepts of guidance and governance. Sustainability transitions are usually associated with a long-term goal that defines the direction of the transition. However, it is crucial to keep in mind that the interpretation of sustainability may vary in time and between actors [32].

Scholars of transition research were able to conceptualize frameworks to properly analyze sustainability transitions such as the Technological Innovation System (TIS), the Transition Management (TM), the Strategic Niche Management (SNM), and the Multi-Level Perspective (MLP) [12,32].

The Multi-Level Perspective (MLP) is an analytical framework that combines ideas from evolutionary economics, sociology of innovation, and institutional theory [12]. This framework suggests that transitions are the result of the interaction of niches, regimes, and landscapes [32]. Niches are considered to be the protected space in which revolutionary ideas emerge, nurtured by pioneers and leading entrepreneurs who experiment with new configurations [33]. These revolutionary innovations can spread widely and change the current regime, but this requires the regime to be under pressure emanating from landscape developments (Figure 1). The MLP is particularly interested in the systemic dimensions of transitions that are reflected by the different degrees of the structuration of each analytical level [12].

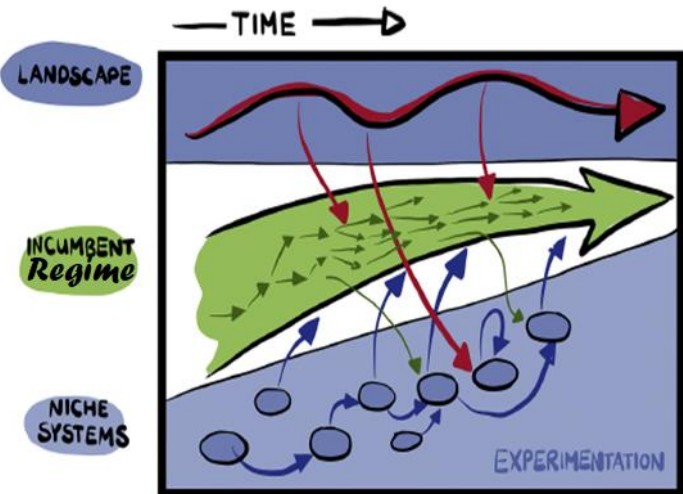

**Figure 1.** Visualization of the MLP and dynamics of the socio-technical regimes. Source: Reprinted with permission from van Rijnsoever et al. [34]. 2020, Elsevier.

Additionally, scholars have identified certain types of transition pathways depending on the niche–regime–landscape interaction [32,35]. Geels and Schot [31] described one of the major typologies, suggesting that the nature of the interaction between niche and landscape developments can be a reinforcing or a disruptive one, and that the niche can be in various stages of development depending on the timing of the transition. Transition

pathways can be differentiated using a combination of different levels of these criteria. This typology describes four major pathways that are *substitution*, *transformation*, *reconfiguration*, and *de-alignment and re-alignment* pathways as well as two additional pathways for the case of a stable regime (*reproduction process*) and that of a combination of pathways (*sequence of transition pathways*).

*2.2. Data Collection and Analysis*

The objective behind this step is to gather relevant scholarly articles regarding the topics of the article. In order to accomplish this, we used multiple research strings to cover the desired topics and to find related articles that present important data about the topics. We then filtered through the articles to find and select those that present valuable data and conclusions in order to include them in our analysis.

A systematic review of the literature was done during the period from the 10th to the 20th of July 2021 on Clarivate Analytics Web of Science databases as well as Google Scholar databases, yielding 301 records, out of which 11 were retained for their relevance to our topic after the screening of the titles and abstracts of the publications (Table 1). The queries used for this review were formulated with keywords that explored different aspects of Moroccan agriculture (e.g., policies, climate change, sustainability transitions), with search fields comprising titles, abstracts, and keywords. The following is a summary of said review.

**Table 1.** Summary of the literature review of the Web of Science database.

| Search Theme | Search String | Number of Records Obtained by the Search | Number of Selected Records after Screening | References of the Selected Records |
|---|---|---|---|---|
| General state of Moroccan organic agriculture | Morocco AND (organic agriculture OR organic farming) | 57 | 1 | Lemeilleur and Sermage [28] |
| Effects of climate change on Moroccan agriculture | Morocco AND agriculture AND climate change | 81 | 2 | Lagrini et al. [9]; Schilling et al. [36] |
| Policies shaping Moroccan agriculture | Morocco AND agriculture AND policies | 80 | 4 | Sraïri [11]; Ouraich and Tyner [37]; Schilling et al. [38]; El Youssfi et al. [5] |
| The state of food security in Morocco | Morocco AND "Food security" | 81 | 4 | Abdelhedi and Zouari [39]; Saidi and Diouri [40]; Onyiriuba et al. [4]; Soffiantini [41] |
| Sustainability transition studies in the Moroccan context | Morocco AND "sustainability transitions" | 2 | 0 | |

Primary data were collected through the use of semi-structured interviews. The objective behind this step was to collect relevant qualitative data from different stakeholders of the Moroccan organic sector, be it producers, processors, or experts. We inquired about the opinions and visions of the interviewees and translated them into significant conclusions that helped us analyze the different topics related to our conceptual framework.

Interviewing is a specific method of data collection; it differs from a normal survey by being more flexible and less strictly reliant on the questionnaire. Interviews are discussions meant to gather information about specific topics. They can be conducted in person or through phone calls, etc. The characteristics of the interviewing method allow us to collect two types of data: personal opinions and background information. In our case, we will be covering both typologies as we inquire about the opinions and attitudes of different

stakeholders regarding the organic transition as well as ask about their knowledge of factors that are affecting the transition [42].

We chose to opt for a semi-structured interview as it ensures coverage of the right topics while granting the interviewee enough freedom to express issues that might not be initially mentioned in the interview guide [42].

The diversity of the interviewees requires a diverse interview guide. In this regard, we opted for the elaboration of a general interview guide in which questions are adapted depending on the stakeholders' affiliations (see Appendix A). The interview guide is composed of four sections. After establishing the participants' consent and informing them of the anonymity of their statements, the participants start responding to the first section A, which is a simple identification questionnaire that allows for a general description of the interviewee. The remaining sections correspond to the main objectives of our research: Section B concerns the external factors that affect the organic sector in Morocco, section C inquires about the general features and characteristics of the organic transition, and, lastly, section D concerns the relations between the organic niche and the conventional regimes. The interviews followed the general guide and depending on the interviewees' responses, we tried to ask more questions about the topics to which they accorded more importance.

The interviews took place in Morocco in the period 29 July until 6 September 2021. The state of affairs in Morocco made it difficult to have the interviews conducted face-to-face due to the ongoing COVID-19 pandemic; this is why we mostly relied on phone communications and email responses to collect data from the interviewees. Thus, we were able to conduct eleven interviews with different actors in the organic sector following a convenience sampling that would allow us to collect data from diverse stakeholders. Interviewees were chosen based on their knowledge of the Moroccan organic sector and the relevance of their activities in the sector.

The interviews yielded 10 audio recordings, ranging from 22 to 51 min, and one written response (Table 2) in Moroccan dialect and French, which were later transcribed and translated into English text.

**Table 2.** List of conducted interviews.

| Interview Date | Interviewee Reference | Duration (min) |
| --- | --- | --- |
| 29 July 2021 | 1_Assoc | 32 |
| 3 August 2021 | 2_Prod | 22 |
| 4 August 2021 | 3_Cert | 39 |
| 8 August 2021 | 4_Prod_Coop | 34 |
| 9 August 2021 | 5_Minis_Assoc_Prod | 51 |
| 9 August 2021 | 6_Minis | 28 |
| 11 August 2021 | 7_Minis | 33 |
| 26 August 2021 | 8_Minis | 49 |
| 31 August 2021 | 9_Prod_Coop | 41 |
| 1 September 2021 | 10_Minis_Coop_Assoc | 34 |
| 6 September 2021 | 11_Minis | Email |

In order to transform the collected interview transcripts into relevant data, we coded the transcripts to easily filter the data into relevant themes for the subsequent analysis through the MLP framework.

Using the Microsoft Word extension "DocTools ExtractData", we attributed codes to different quotes of the transcripts, starting with codes that were concept-driven (or inductive, "producers", "consumers", and "environment challenges", for example) which are codes whose relevance stems from the literature review [21,27,43]. Afterwards, data-

driven codes (or deductive) were added to the list of codes because of their relevance to the interviewees [44].

Finally, the coded quotes were extracted and rearranged into an Excel sheet to facilitate filtering and searching of quotes by different filters. Taking inspiration from the thematic analysis method, we grouped the codes into themes and subthemes that regroup similar and inter-dependent codes to give a summarized idea of the actors' thoughts and opinions regarding those specific themes [44].

The choice of the themes was based on the subsequent analysis framework we are using. We opted for a distinction between the themes of "niche", "regime", and "landscape" to facilitate the analysis of the resulting data (Figure 2). Some of the themes chosen entailed distinct sub-themes that needed to be addressed specifically. The codes linked to each theme and sub-theme varied in accuracy; some codes were linked directly to the sub-theme they belonged to (for example, "external market" representing quotes exclusively linked to the sub-theme of the same name), while other codes were relatively vaguer and could be used in multiple themes (codes such as "price", for example) [45].

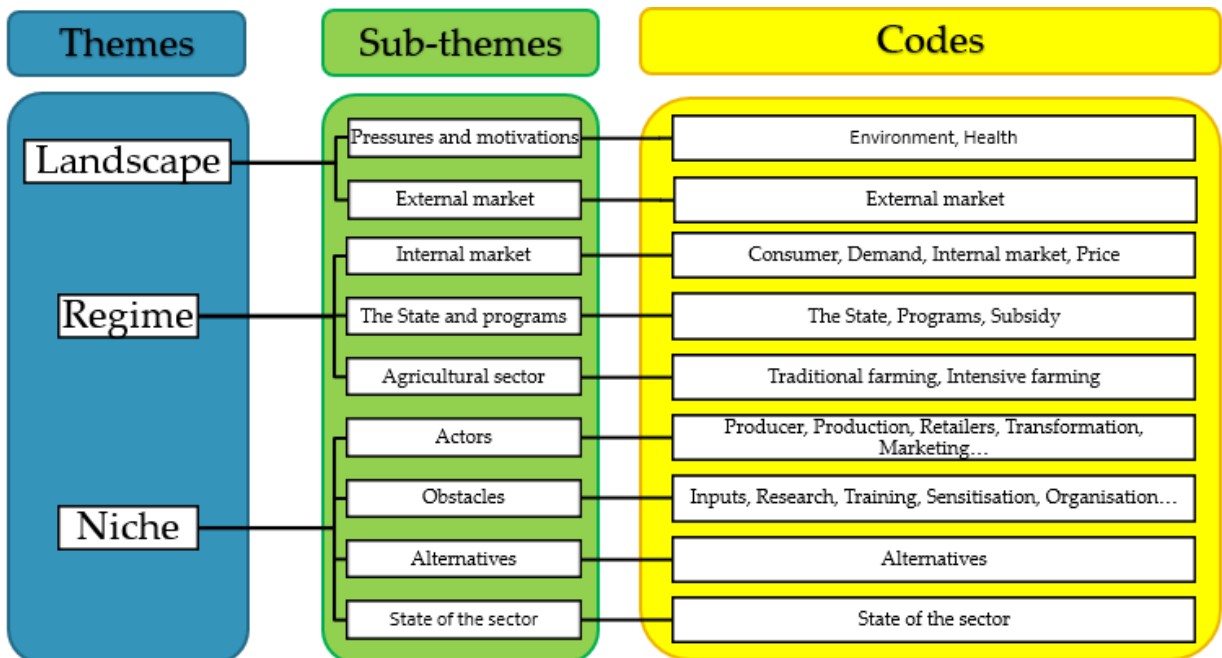

**Figure 2.** Themes and sub-themes used in thematic analysis. Source: Authors' elaboration.

## 3. Results

In this part, we will explore the internal dynamics and different interactions of each socio-technical level of the agri-food system surrounding the organic niche.

### 3.1. Landscape Level Dynamics

The Moroccan agricultural sector consists of 1.5 million farms, 70% of which have an area of less than five ha. For 55% of them, this figure falls below three ha. The latter only exploit 12% of the total arable area, while 1% of farms have more than 50 ha and cultivate 15% of arable lands, clearly showing the divide that exists in the conventional agriculture regime between small farms in rural settings and bigger industrial farms [10,39]. Thus, the majority of farms are too small to be able to mobilize the technical and financial resources needed for agricultural intensification. Additionally, given the importance of non-irrigated lands in the total arable area, these farms face a strong dependence on climatic hazards and particularly on very irregular rainfall [9,21].

As a North African country, Morocco deals with major environmental setbacks regarding water and soil resources. These challenges are the natural result of the country's

climate and geography, but they are emphasized by the economic growth of the country leading to more urbanization while simultaneously having limited funds and frameworks for resource management. This puts around 15 million hectares of land under threat of degradation due to lower fertility rates as well as reducing water reservoir capacities through siltation mechanisms [1]. Additionally, Morocco is considered to be one of the most sensitive countries towards climate change due to rain-fed agriculture representing more than 90% of the production [5,37]. Some of the direct effects of climate change in Morocco are the shift in rainfall patterns that causes yield losses, and the recurring droughts that affect social stability and accentuate the existing inequality between farmers [36,38].

Another pressuring factor for the agri-food system is the Moroccan diet, which relies on cereals to provide 60% of food energy supplies with an average consumption of 200 kg/year/capita of cereals [40]. Cereal production being highly dependent on rainfall makes it difficult to achieve self-sufficiency, therefore Morocco has to import up to 40% of its national needs of cereals [5]. The dependence on the external market goes beyond imports and includes exports as well. Moroccan policies encouraged more crop production via the Green Morocco Plan (2008–2018) and did succeed in increasing the yields of many crops, the downside being the low efforts put into commercialization resulting in difficulties in exporting excess production [4,5,37]. Thus, Morocco is depending on the international market to import crucial staple foods and to export high added value crops such as tomatoes and citrus. Unfortunately, this exchange is unfavorable for Morocco as fruits and vegetables require more water to grow than cereals, thus increasing Morocco's sensitivity to water shortage even further [11,40].

While the landscape level clearly pressures the conventional agriculture regime, it does not necessarily affect the niches in the same way. A portion of Moroccan consumers has been influenced by the rising awareness regarding health and nutrition. These consumer needs for healthy products encouraged farmers to explore organic farming [28].

It is true that Morocco is vulnerable to environmental challenges; however, the country's diverse climates and soils present it with an opportunity to grow multiple crops at different times of the year. These conditions also permit Moroccan farmers to cultivate specific crops that are difficult to manage in European climatic conditions. These factors are especially encouraging for organic farming as it translates into an off-season production of special products that are valuable to the external market [21].

### 3.2. Regime Level Dynamics

While we showed that climate change threatens Morocco's agriculture, it is important to signal that Moroccan policies and agricultural practices also contribute to the degradation of environmental resources; in fact, environmental degradation cost around 3.5% of the gross domestic product (MAD 32.5 billion) in 2014 [46]. Over the last decades, Morocco's economic growth led to a significant increase in greenhouse gas emissions, 21% of which are linked to agriculture [46]. Moroccan agriculture started becoming more intensive since the middle of the 20th century, with the adoption of nitrogen fertilization, new irrigation systems, and the introduction of mechanization [37,38]. Pesticide use has also been proven to cause biodiversity loss and health issues, especially with inadequate use by farmers [47].

The history of Moroccan rural areas and public policies since the period of the protectorate have resulted in notable diversity and a tradition/modernity dualism. The State's efforts to reduce the major economic inequalities between the two poles of agriculture are reflected in major strategies and development programs. These programs adopt a partnership approach with groups of producers or rural people. However, the dynamics of collective action do not follow the efforts deployed. On the one hand, the populations have difficulty in discarding the strong past of State intervention, and on the other hand, the accelerated rate of the creation of collective structures (cooperatives, associations, etc.) does not allow farmers to assimilate the model [6].

Although economically insignificant, this type of farming role is essential, as it serves as a "social anchor" by providing the households concerned with the security of an at-

tachment to rural society. It thus contributes to limiting the rural exodus and the anarchic growth of peri-urban areas, a source of instability for the country [10,39].

The interviewees also signaled the existing duality in the Moroccan agricultural system. Additionally, they stated that the small farmers were generally using less inputs and carried on traditional practices that were considered environment friendly. Thus, this typology of farmers was considered to be close to organic farming principles and presented a good opportunity for conversion.

The second type of industrial farms that are highly reliant on inputs face more challenges in the conversion process and tend to be less motivated to convert because of the perceived lower income of organic farming.

### 3.2.1. Consumers and Internal Market

The market for organic products is developing at a rather timid pace. The deficit of communication surrounding these products hinders their integration into the local market and they struggle to gain the confidence of the Moroccan consumer, who is unaware of the real virtues of this category of products. The solution to this situation lies in the establishment of an information platform allowing consumers to orient themselves towards organic products and to have an answer to all their questions to adopt a healthy lifestyle [28,48].

According to the interviewees, consumers are gradually becoming more aware of the importance of nutrition in leading a healthy lifestyle. This means that a consumer base that appreciates organic products is forming in the Moroccan context. However, organic product tendencies to be costly worsens the marketing opportunity because of Moroccan consumers' low purchasing capacity. Additionally, the prevalence of traditional farming, commonly called "Beldi", tends to confuse Moroccan consumers, since they are generally misinformed about organic production standards and cannot differentiate between organic and "Beldi" products. The interviews indicated that the internal demand for organic products is insufficient for the moment, thus more sensitization and encouragement of consumers is needed.

The Moroccan consumer can possibly pay more to consume organic, but on the other hand, he asks that the products be diversified and available all year round with an assurance of quality and continuous traceability by the competent services at the national market level [49].

### 3.2.2. The State and Its Programs

Agriculture has long held a central place in the country's growth model and has benefited from significant support, however its performance is deemed less than optimal. The successive agricultural policies of the last 50 years have brought undeniable progress, but have not been able to address agricultural development in its entirety, in its diversity, and in its fundamental relationship with rural development. They have remained overly technical and directive and have not been able to address human capital or mobilize and empower actors or effectively support their initiatives. State intervention models have contributed to an impoverished vision of agricultural development, largely inhibiting a potential for innovation based on the richness of diversity, on the mobilization of actors, and the adaptation to market opportunities. Rural development lag has accumulated and an important cash crop agriculture has been created [10].

Thus, until the end of the 2000s, while shifting from an interventionist policy of the State in the sector (support for production prices of strategic products as part of its policy of self-sufficiency and support for production to replace imports) to a more liberal policy (replacing of support for production prices by support for agricultural investment), large-scale strategic actions have been undertaken, such as the policy of dams, the establishment of hydro-agricultural infrastructure, the strengthening of plant and animal production, and the supervision of farmers.

These efforts have enabled the sector to make several achievements, in particular the construction of hydro-agricultural infrastructure, the diversification of crop and animal

production, the capitalization of recognized ancestral know-how (cultivation and irrigation practices, etc.), and the development of the fruit and vegetable sector [2].

Nevertheless, certain constraints remained and hindered the effectiveness of the Moroccan agricultural development model. These constraints related, in particular, to a governance deficit (inefficiency of public interventions with institutional weakness at the territorial level, weaknesses in terms of public–private partnerships and professional organization, increased centralization, etc.), inadequate treatment of the problems of the agricultural sector and a lack of transparency, inadequate treatment of the land issues hindering the development of agricultural investment (knowing that smallholders represent 70% of farms), insufficiently prepared human capital that is unable to contribute to the modernization of the sector (high illiteracy rate, low adoption of technologies), insufficiently rationalized management of water resources (low efficiency of plot irrigation dominated by gravity systems), as well as the weak organization of certain sectors [2,11].

Interviewees also affirmed that the State held the most influence on the development of the organic sector. They also emphasized the importance of cooperation between the State and the operators for the sector to develop. According to the interviewees, it is this lack of cooperation and organization that caused the previous organic support programs to fall short of their objectives. An example of these shortcomings is the 2011 contract program between the ministry and the FIMABIO federation [21,50].

Additionally, many actors think that the State programs failed to support operators properly because of the inadequacy of the subsidies and the unavailability of research and training programs.

Other actors think that State programs hinder the development of the organic sector by encouraging the intensification of conventional agriculture. State programs have made chemical inputs and conventional practices popular and easily available, which lessens the popularity of organic farming. In this regard, one of the interviewees stated: *"What is needed is a specific agency in the ministry dedicated to the organic sector. One that provides consulting and guidance. The current national consulting offices are clueless about organic agriculture and still recommend chemical inputs for farmers"*.

### 3.3. Niche Level Dynamics

In this section, we will explore how the organic niche came to be in the Moroccan context, the dynamics of the niche through its actors and the challenges they face, as well as the other niches and alternative production systems that exist at this socio-technical level.

Organic farming was introduced in Morocco in 1986 through private operators growing olives and citrus in the Marrakech and Benslimane regions, respectively. However, the organic sector did not truly become relevant until 2011 when the first program promoting organic agriculture was signed between the government and the Moroccan association of the organic production sector (AMABIO). The objectives of the program were to expand the organically cultivated land to reach 40,000 ha, to increase the organic production to 400,000 T, to create the equivalent of 35,000 job opportunities, to increase the national consumption of organic products, and to put in place legislation for organic agriculture within 9 years. The total investment foreseen for these actions was MAD 1121 billion, of which MAD 286 million was covered by the State while the rest was covered by the inter-professional federation of the organic sector (FIMABIO) [28,50].

Interviewees also told the same story of international demand encouraging organic farming to make its entrance into the Moroccan context in the form of amateur farmers who sought to export their products. Afterwards, the State showed its interest in organic farming as a way to valorize production. Organic farming became subject to a contract program in 2011 to recruit more operators and to expand the organic farming area. While the objectives were not met, the creation of the FIMABIO federation and the creation of the organic legislation were important steps in the development of organic farming.

Eventually, the internal demand for organic products started to take shape, although its relevance compared with external demand is debatable. Moroccan consumers are indeed

becoming more aware of the importance of healthy nutrition, thus some of them are starting to seek high-quality food regardless of the price. However, these consumers are considered a minority. Interviewees also emphasized the importance of economic profit as a driver for the adoption of organic farming via its premium price and exportation potential.

Currently, Morocco has an area of more than 9500 ha of organic certified cultivated farmland, 980 ha of farmland in transition, 273,000 ha of wild collection areas, and a total organic production of 104,600 T in 2018 [50]. The surveys clearly show that the organic niche in Morocco is not progressing as planned by the 2011 program.

Some of the reasons behind this slow progress are the absence of adequate infrastructures dedicated to the handling and commercialization of products on the internal market, the unavailability of qualified technicians in the organic sector, and the shortage of subsidies for production, export, and processing [49].

### 3.3.1. Actors of the Niche

The Moroccan organic sector regroups 302 operators. Producers represent the majority, while transformers are very few and mostly consist of farmers who engage in the packaging, trituration, or drying of their own products [49]. Research institutes also show a low level of involvement in the sector. Throughout the interviews, this situation was made more apparent by the interviewees' accounts. They stated that indeed the transformation and research actors are few or completely inexistent.

Organic farmers, similar to conventional ones, adhere to the typology of big and small farmers. In addition to that, their motivations to practice organic farming also vary. Most interviewed actors stated that farmers are generally driven by economic profit to convert, since organic products are generally costly and present the opportunity of being exported. Other interviewees mentioned that there are farmers who are driven by their convictions about leading healthy lifestyles and respect for the environment to convert to organic.

This distinction of motivation translates to a difference in the level of involvement and knowledge about organic farming. Farmers that are motivated by profit tend to find the conversion period challenging and often resort to inputs at the slightest inconvenience. In opposition to that, farmers who are adhering to organic principles by conviction tend to be more knowledgeable about organic practices and tolerate a certain level of economic loss, since they believe that organic product value goes beyond their revenue and includes benefits to their health and the health of their soil.

Organization-wise, the operators of the organic sector created the FIMABIO federation in 2016, which consisted of producers, transformers, and retailers. By 2017, they were able to obtain recognition from the State and the FIMABIO was considered the representative of operators at a national level. However, this recognition was revoked from the federation in December 2020, leaving the sector with no clear representation [51]. Interviews revealed that the reason behind this revoking is the federation's inability to achieve the goals set by the contract programs. This was mainly due to conflicting interests and the uncooperativeness of producers and retailers, who sought different goals and ultimately jeopardized the organization of the sector.

### 3.3.2. Obstacles Facing the Niche

Throughout the interviews, it was made clear that production was the most important theme discussed by the participants, many of whom believed that it was key to the development of the organic agriculture sector and thus elaborated on many aspects that affected the production aspect.

Organic farming is notorious for its demanding production standards; the certification process as well as the conversion period are challenging steps that discourage many farmers from adopting organic farming. The prohibition of chemical inputs is also a relevant problem for Moroccan farmers as they are easily available and their use is encouraged by many State programs and firms.

In general, Moroccan farmers are uninformed about organic farming. Their high illiteracy rates and the lack of training and sensitization around the topic of organic farming make it difficult to introduce them to the organic sector, a sector that typically requires a high level of knowledge and implication to adhere to. Thus, the number of farmers who show interest in organic farming remains low.

Interviewees have stated that organic inputs are a major obstacle to the production process. Organic inputs are mainly sourced through importation. Currently, there are very few producers of organic seed and organic fertilizers and pesticides. This adds another layer of difficulty to farmers practicing organic farming. Additionally, producers generally lack knowledge and expertise to adopt organic farming, especially in the first years of conversion when they cannot market their products as organic but still have to apply the organic standards.

Different actors have emphasized the importance of certification as a defining part of the sector; it is what establishes the difference between organic farming and the other existing alternatives and thus is considered to be the key process that valorizes products and adds value to them. However, the certification process is notorious for being costly and complex for farmers in general.

These challenges do not affect all Moroccan farmers the same way, rather there are factors that can ease the certification process or mitigate the unavailability of organic inputs. In general, producers lack the knowledge and expertise to adopt organic farming, especially in the first years of conversion. They are nonetheless motivated to do so to improve their income by accessing external markets. On the other hand, farmers that are pursuing organic farming because of their convictions and beliefs find it easier to adapt to the challenges. These farmers, who are usually familiar with organic practices or similar sustainable practices such as agro-ecology, are less affected by the input shortage as they have more knowledge and solutions to circumvent the need for inputs. Small traditional farmers are similar to them as well since they do not use inputs at all generally, while intensive farms are generally harder to convert due to their over-reliance on chemical inputs.

The type of crops is also a factor in the conversion process. Spontaneous low-input crops are easily converted compared with demanding and sensitive crops.

Based on these differences, interviewees have emphasized that the niche's true potential lies within these small motivated farmers and the wide natural collection areas that are easy to convert to organic but that also benefit the most from the conversion.

In order to alleviate the certification challenges, the State did allocate a specific subsidy for the costs of the certification. Unfortunately, interviewees stated that this subsidy had a limited effect in encouraging farmers. The reason given was the complexity of the paperwork behind it, which made it difficult to obtain, given the fact that most farmers thought it was not worth the effort or simply could not fulfil its bureaucratic requirements.

As we mentioned before, transformation is severely undeveloped in the Moroccan organic sector. The reason behind this is the atomization of the supply of organic raw products, which does not guarantee processors a regular supply. This constraint is reinforced by the obligation to meet traceability requirements during the transformation process, notably with the separation in time and/or space of organic and non-organic productions [49].

Generally, Moroccan organic products are mainly meant for exportation. However, national demand has been growing for several years, with the appearance of numerous distribution channels for organic products in the internal market. Large and medium-sized supermarkets, specialized stores, and direct sales on farms are the most common sale points for organic products [49]. Despite this trend, the Moroccan consumer is still unknowledgeable of organic products and their consumption remains low on the internal market [48], and organic products remain challenging to market due to their seasonal nature and low production volume, which is the opposite of what retailers and consumers generally prefer.

Interviews revealed the same phenomenon. Consumers are indeed starting to show interest in organic products, but it is the highly educated and wealthy consumers who do

so. Some interviewees criticized this aspect as they believed organic products should be "democratized" across all consumers. Other interviewees accepted the elitist nature of the niche and argued that it is normal for a developing sector to start this way.

Interviewees signaled the importance of sensitization for consumers to introduce them to organic products. This is especially important as Moroccan consumers already appreciate 'Beldi' or traditional farming products, thus they would easily make a positive association between organic and 'Beldi' products if they understood the similarities and added value that organic products have.

We mentioned earlier that the organic sector in Morocco lacks organization. This aspect is negatively impacting all sub-sectors from production to marketing. Interviewees have signaled this and added that many of the obstacles of the sector could be overcome if organization were not an issue. The importance of organization lies within the recognition of the State and the elaboration of meaningful development programs. One of the interviewees stated: "*The revoking of the Federation's status as representative definitely slowed the development of the sector. Now we feel as if the sector is on standby while the ministry decides to either bypass any form of farmers' association, or wait and form a new representative for the farmers*".

Additionally, a well-organized sector can easily diffuse information that would help producers overcome difficulties and develop a sense of solidarity between them. Organization across different actors can also help create additional added value and facilitate the creation of profitable value chains between producers, processors, and retailers.

### 3.3.3. Similar Alternatives

Organic farming is not the only alternative production system that exists in the Moroccan context. Interviews mentioned the existence of other similar initiatives such as agro-ecology, permaculture, conservation agriculture, and protected geographic indications. These alternatives were deemed similar to organic farming in regard to their aim of valorizing products, protecting environmental resources, and providing healthy nutrition.

However, interviewees had different opinions about their legitimacy. The fact that these alternatives generally lack a precise production standard and have little to no legislation supporting them greatly decreases their legitimacy. Interviewees argued that the existence of such unregulated production methods greatly undermines the value of organic products that require rigorous standards and certification. Additionally, these alternatives are also competing for the State's support, resulting in a decrease of State allocated funds towards organic farming. Furthermore, the existence of multiple labels might confuse the customer and reflect negatively on the consumer's trust in the organic label.

Other interviewees thought that these initiatives can work hand in hand with organic farming and act as training grounds that farmers can adopt before fully engaging in proper organic farming. This is due to the shared practices that these alternatives have, such as the prohibition of chemical inputs, the attention accorded to soil health, and their aim to preserve endemic species and develop them.

## 4. Discussion

### 4.1. Landscape Pressures

External pressures are crucial in understanding sustainability transitions. They are the factors pushing regime actors to acknowledge the need for change and to adapt to it.

Based on the prior descriptions, the Moroccan agri-food regime faces multiple pressures emanating from the landscape. Environmental challenges are constantly threatening the present and future productivity, and as a consequence they stir social unrest, especially considering the proportion of the population relying on agriculture for their income. Additionally, the regime actors are influenced by the general ongoing trends that call for more sustainable production patterns and healthier lifestyles. Finally, Morocco's reliance on exports to sell agricultural products and imports to achieve food security is a significant challenge that constantly affects the country.

The pressure presented by these factors is not focused on a singular moment in time, but rather it is a "disruptive" pressure. The regime actors are not compelled to address environmental pressures since they are considered grand challenges that require a long-term commitment. The country's dependence on external markets has always been the case and the regime can manage this situation through multiple programs. A clear indication of this success is the country's stability during the food insecurity induced "Arab Spring" riots [41]. This is a strong indicator of the regime's ability to withstand specific shocks emanating from the landscape.

In other words, the landscape poses little pressure in the short term, however the pressures are continuously present and building up to become more imposing in the long term. As resources are depleted, climate change is accentuated, and social inequalities grow wider [31].

*4.2. Niche Interactions with the Regime*

The concept of anchoring will be of good use in exploring the interactions occurring between the niche and regime. Anchoring refers to the newly formed connections between the regime and the niche. The more connections that are formed, the higher the chances for the adoption of said niche [52]. We will distinguish different types of anchoring by defining the particular domain to which organic farming connects in the regime, which can be either the technological domain, the institutional domain, or the network one.

In the context of the occurring environmental challenges, Moroccan policies shifted towards more sustainable alternatives that preserve non-renewable resources by adopting newer systems of irrigation, encouraging ecologically friendly cropping methods, such as conservation agriculture and organic farming, and valorizing agricultural products through labels and production standards [6]. Thus, there was a significant anchoring of organic farming on the normative institutional level of the regime. The regime has adopted organic legislation and integrated organic farming as a way to valorize agricultural products and to improve small farmers' livelihood in its programs. Although these programs tend to fall short of achieving their objectives, their existence is a clear indication of the linkage that is happening on the institutional level.

Secondly, there was a temporary link between the regime and the niche's network when the FIMABIO federation benefitted from the recognition of the State. This network anchoring turned out to be temporary, as the State later revoked its recognition because of the federation's inability to cooperate effectively. This goes to show how anchoring is not a permanent process in the context of newly emerging niches but rather a volatile one.

Organic farming benefits from the ideas of consumers and farmers about traditional farming to gain popularity and appeal. The similarity between organic and traditional farming makes it easier to adopt organic farming as it relates to the Moroccan values of simplicity and respect for traditions. Thus, organic agriculture easily relates to the institutional level of the regime, specifically the cognitive one that represents people's social values and interests [52].

Finally, organic farming also anchors itself through the existing traditional farming practices that are also common to organic farming such as "the low use of pesticides in multiple crops and a high integration between crop production and livestock which satisfies the farmer's need for manure" [43].

*4.3. Type of Transition Pathway*

Defining which pathway the organic niche is more likely to follow requires having an idea about the nature of the niche–regime and regime–landscape interactions as well as finding the timing or stage at which the niche is currently in accordance with the typology of Geels and Schot [31].

Based on the elements from previous sections, we can conclude that the niche is synergized with the regime. This is especially true due to the integration of organic farming

in multiple programs elaborated by the State, which is considered to be the most influential actor in the regime.

As we have already established, the landscape is pressuring the regime in a disruptive and continuous way, meaning that these pressures are more long-term focused and less relevant in the short term.

The timing or stage of the niche relates to the development status of the niche. Both the literature and interviewees' opinions agree that organic farming is considered an undeveloped niche. While the sector does hold significant potential for growth, statistics and stakeholders' opinions show that there are many factors yet to develop within the sector.

Therefore, the possible pathways with regard to the timing and nature taken by the transition in question can be either the *transformation pathway*, the *reconfiguration pathway*, or a *sequence of transition pathways*.

The *transformation pathway* is especially interesting for our context as it occurs when "Moderate landscape pressure occurs early in disruptive landscape change. Niche-innovations cannot take advantage [. . . ] because they are not sufficiently developed" [31]. The initially moderate pressure can be difficult to perceive from within the regime, which is why it is important to have outsiders draw attention to these pressures. In our context, worldwide experts are the actors drawing the Moroccan regime's attention to environmental pressures. Additionally, outsiders can demonstrate promising alternatives and inspire the regime actors to adopt them, which is precisely the case for the organic niche as an external innovation that the Moroccan regime is starting to adopt. The *transformation pathway* achieves transition through cumulative reorientations by the niche actors who adjust themselves and pressure the regime to change its rules. Through this process, the actors of the regime might remain the same, but the social network they belong to might change.

The *reconfiguration pathway* is a similar case to *transformation pathway*, where the niche innovations are symbiotic with the regime and are easily adopted. The difference between the two is that, while the *transformation pathway* does not necessarily cause deep changes in the regime's architecture, the *reconfiguration pathway* causes deep changes when the adopted innovations keep causing further changes requiring further adjustments, to a point where other niche innovations are introduced to the regime as the regime actors explore new possibilities. This is especially true in our case, since we are dealing with the agri-food system, which is known for having multiple interlaying subsystems where innovations are likely to add up after a major regime change. In this type of pathway, the transition is the result of multiple niche innovations that break through around the initial innovation that sparked the change.

A *sequence of transition pathways* applies in the event where the landscape pressure begins by being moderately disruptive but later becomes significantly pronounced, as in climate change posing environmental pressure that causes civil unrest or a deep economic setback. Regime actors start perceiving moderate change at first and the system begins the transition under a *transformation pathway*. Later on, as the pressure increases, more changes are encouraged and the transition becomes a *reconfiguration*. Finally, when the pressure becomes too high, the regime "collapses" and many niche innovations fill the "vacuum" until eventually one of them resurges as the new regime, in what we call a *de-alignment–re-alignment pathway*. Alternatively, if the pressures happen at a timing when a niche-innovation is well developed, a *substitution* occurs where the innovation replaces the old regime.

## 5. Conclusions

This article contributes to filling the research gap that exists in the field of sustainability transitions of agri-food systems in developing countries by analyzing the dynamics and development process of the organic agriculture niche in Morocco, a North African developing country. In fact, the article analyzes the type and features related to the pathway of the transition to organic agriculture in the country; defines the environmental components and factors that affect the organic transition and analyzes the interrelations between them; and

describes the relations among the organic agriculture niche, the socio-technical landscape, and the other existing socio-technical regimes within the Moroccan agri-food system. The analysis is carried out through the lens of the MLP and combines secondary data from the scholarly and the grey literature as well as primary data from semi-structured interviews with key actors in the Moroccan organic farming sector.

The analysis shows that the current state of the Moroccan agri-food system has proven to be lacking in sustainability, be it environmental, economic, or social sustainability. As it stands now, Morocco's agriculture is depleting non-renewable resources, such as soil and water, and causing social unrest because of its dependence on a fluctuating climate without being able to reach self-sufficiency. This situation calls for a meaningful sustainability transition guided by efficient and effective long-term government programs and carried out by the actors of the current regime.

As we have shown in this article, organic farming can be a reliable alternative to alleviate the effects of intensive farming and to enhance the performance of the agri-food system sustainability-wise. The organic niche in Morocco has a great potential to achieve this, since it benefits from governmental support alongside favorable climatic conditions and a growing interest in agro-ecology. However, the organic niche is still struggling to take-off for several reasons mostly related to organization and awareness of the value of organic farming. Throughout this article, we have shown how the organic niche benefits quite efficiently from the pressures that the landscape is putting on the Moroccan agri-food regime, and we suggested pathways that this transition might take based on the organic niche's synergy with regime components (cf. inclusion in State programs and proximity to traditional farming), the disruptive nature of the environmental challenges posed by the landscape, and the undeveloped stage in which the organic niche is in right now in Morocco.

Other studies and reports in this context have also stated the embryonic stage where the organic sector is currently situated. However, to our knowledge, this is the first study that uses the sustainability transition's framework, namely the Multi-Level Perspective, to explore the state of the Moroccan organic sector and the different dynamics that shape its transition.

The present study suggests that, depending on the nature and the timing of the interaction between the elements of the Moroccan agri-food system, the transition to organic agriculture in Morocco can follow different pathways, viz. *transformation pathway*, *reconfiguration pathway*, or a *sequence of transition pathways*. Future studies are needed to refine this typology by a deeper analysis of the trajectory of the organic sector development in the country and by meticulous scrutiny of future scenarios. What is clear is that the development trajectory, and consequently the transition pathway, will depend to a large extent on public policies as well as the awareness of Moroccan consumers that will shape the domestic market.

Our study used the lens of the Multi-Level Perspective as the central framework piece through which the collected primary and secondary data were analyzed. Evidently, there are merits and criticisms for the use of the MLP in this context, thus we can recommend exploring other sustainability transition frameworks to be applied to the analysis of the organic niche to build a more complete picture and to patch whatever holes the MLP might have left. Additionally, our methodology included the collection of primary data using a semi-structured interview with broad topics. The broad nature of the interviews was a necessity since the study is considered an exploratory one; thus, future research can use the present article as a baseline to precisely frame the topics of data collection and to target key actors with adequate questions.

**Author Contributions:** Conceptualization, H.E.G., R.H., and H.E.B.; methodology, H.E.G., R.H., and H.E.B.; software, H.E.G.; validation, H.E.G., R.H., and H.E.B.; formal analysis, H.E.B., R.H., and H.E.B.; investigation, H.E.G.; data curation, H.E.G.; writing—original draft preparation, H.E.G., R.H., and H.E.B.; writing—review and editing, H.E.G., R.H., and H.E.B.; visualization, H.E.G.; supervision,

R.H., and H.E.B.; project administration, H.E.B. All authors have read and agreed to the published version of the manuscript.

**Funding:** This research received no external funding.

**Institutional Review Board Statement:** Not applicable.

**Informed Consent Statement:** Not applicable.

**Data Availability Statement:** Not applicable.

**Acknowledgments:** We thank our colleagues from Hassan II Institute of Agronomy and Veterinary Sciences, and the International Centre for Advanced Mediterranean Agronomic Studies (CIHEAM-Bari), who provided insight and expertise that greatly assisted the research. We would also like to thank the entirety of the interviewees for their cooperation with data collection and for providing us with comments that greatly improved the manuscript.

**Conflicts of Interest:** The authors declare no conflict of interest.

## Appendix A. General Interview Guide

### Introduction:

The following is a semi-structured interview, with the objective of collecting data about opinions and visions of different stakeholders of the Moroccan organic sector.

By participating in this interview, you consent for the use of the answers you supply in various types of publications. The details provided are all confidential, and the fact that this instrument is anonymized (i.e., it contains no names or personal addresses) means that no details can be traced back to any respondent.

**A. Interviewee identification:**

1. Gender;
2. Age;
3. Affiliation/occupation

**B. Environment and external factors affecting Moroccan organic agriculture:**

4. What are the factors that help organic agriculture establish itself in the Moroccan agri-food sector?
5. Which sub-sector of organic agriculture (production, processing, marketing, research, etc.) do you think is more crucial?
6. Who are the actors that have a significant influence on organic agriculture in Morocco? Who are the most impacted ones?
7. Can you recall any events that helped/hindered the development of organic agriculture in Morocco?
8. Do you think organic agriculture in Morocco is highly dependent on a specific factor (inputs, legislation, awareness)? Can you explain why?

**C. Organic agriculture transition features:**

9. Are there any movements/sectors that are similar to organic agriculture in Morocco? What do they have in common? Where do they differ?
10. Do you think the organic sector is well established? Do you think that organic stakeholders have the means and potential to develop the sector further?
11. Did any major events (or global tendencies) justify the need for organic agriculture? Is organic agriculture the only answer to these pressures?

**D. Organic agriculture niche relations with the conventional regimes:**

12. Does organic agriculture contradict any existing values/practices of the Moroccan agri-food sector? Are these contradictions unavoidable or can there be compromise?
13. Are there any elements of the organic agriculture standards and principles that are already present in the Moroccan context?

14. How easy is the implementation of organic agriculture in the Moroccan context? Are there any obstacles regarding legislation? Customer behaviors? Input availability?
15. Are there any actors who would be against the development of the organic sector in Morocco? Why is that? Are they negatively affecting organic agriculture's development? How so?

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
