# Peer review of "Pathways of Transition to Organic Agriculture in Morocco"

_world, doi:10.3390/world3030040_

Round 1

Reviewer 1 Report

The work is ok and keep it up for next study.

Author Response

We thank very much the reviewer for the positive evaluation and appreciation of the manuscript.

Reviewer 2 Report

The subject of the article is interesting, and it is linked to the objectives of the journal, however, there are some issues that have to be reconsidered.

For better visibility on databases, the authors are asked not to repeat among keywords the words/concepts included in the title of the article.

The abstract part looks clear, in terms of research importance, objectives, methodology ad results.

The Methodology is quite complicated presented, so difficult to follow. it is recommended to make it more concise, clearer =and easy to replicate. 

A deeper explanation of why the sample is representative for the entire population could be helpful.

The Introduction part tries to describe the domain, while the literature review (results of similar studies etc.) is quite superficially discussed, and research questions are not clearly presented. Both parts should be reconsidered. Please update the references (e.g. https://doi.org/10.3390/su70810521, https://doi.org/10.3390/agriculture11111050 etc.). In Introduction the hypothesis/research objectives are not clearly described.

The results are interesting and they are well discussed, but the conclusions are not enough to sustain the results. The use of the research is, so, insufficiently explained at the conclusion part. It is advisable to create a distinct part for formulating general conclusions and recommendations for scholars, government, businesses etc.

Author Response

We would like to thank the reviewer for the valuable comments and insightful suggestions. We explain in the attached cover letter point-by-point the details of the revisions made in the manuscript and we provide our feedback on all comments. All revisions are easily visible and clearly highlighted in the revised manuscript in track changes.

Reviewer 3 Report

Many thanks to the editors for the invitation. I have read your work carefully.. Specific comments are as follows.

-The abstract should briefly describe the research background and policy implications.

- Research gaps should be well mentioned in the introduction. A good research gap can give the reader more insight.

-Some of the most recent literature (last three years) should be considered and updated.

The following papers can be good examples to help you improve your paper:

-Qing, L.; Chun, D.; Ock, Y.-S.; Dagestani, A.A.; Ma, X. What Myths about Green Technology Innovation and Financial Performance's Relationship? A Bibliometric Analysis Review. Economies 2022, 10, 92. https://doi.org/10.3390/economies10040092

-Muhammad Sadiq, Fenghua Wen, Abd Alwahed Dagestani, Environmental footprint impacts of nuclear energy consumption: The role of environmental technology and globalization in ten largest ecological footprint countries, Nuclear Engineering and Technology,2022,https://doi.org/10.1016/j.net.2022.05.016.

-Qing, L.; Chun, D.; Dagestani, A.A.; Li, P. Does Proactive Green Technology Innovation Improve Financial Performance? Evidence from Listed Companies with Semiconductor Concepts Stock in China. Sustainability 2022, 14, 4600. https://doi.org/10.3390/su14084600

-Dagestani, A. A. (2022). An Analysis of the Impacts of COVID-19 and Freight Cost on Trade of the Economic Belt and the Maritime Silk Road. International Journal of Industrial Engineering & Production Research, 33(3), 1-16.

"the manuscript still has a point not very clear, I hope they may explain, what's your contribution to the theory ? you can choose the theory you think it fits better these articles can be good examples could help you to understand what do I mean by "contribution to the theory" Eesley, C., Li, J. B., & Yang, D. (2016). Does institutional change in universities influence high-tech entrepreneurship? Evidence from China's Project 985. Organization Science, 27(2), 446-461. https://doi.org/10.1287/orsc.2015.1038

-there are some language mistakes and some hard-to-read sentences in the manuscript please do a careful proofreading
regards, go ahead

Author Response

(The authors gave the same response as above.)

Round 2

Reviewer 2 Report

The authors succeded in answering my concerns, the article could be published.